# Association Between Frequency of Away-From-Home Meals and Prevalence of Inflammatory Sinonasal Diseases: A Population-Based Cross-Sectional Study

**DOI:** 10.3390/diseases12110286

**Published:** 2024-11-08

**Authors:** Munsoo Han, Yujin Jeong, Sooun Kwak, Jaemin Shin, Taehoon Kim

**Affiliations:** 1Department of Otorhinolaryngology-Head and Neck Surgery, College of Medicine, Korea University, Seoul 02841, Republic of Korea; mshan35@gmail.com (M.H.); kwaksooun@naver.com (S.K.); shinjm0601@hanmail.net (J.S.); 2Mucosal Immunology Institute, College of Medicine, Korea University, Seoul 02841, Republic of Korea; 3Department of Biostatistics, College of Medicine, Korea University, Seoul 02841, Republic of Korea; wjddbwls1020@korea.ac.kr

**Keywords:** diet, allergic rhinitis, chronic rhinosinusitis, immunoglobulin E, olfaction

## Abstract

At the beginning of the COVID-19 pandemic, people had to stay at home due to quarantine, and the food delivery industry grew significantly. Concerns have been raised regarding the popularity of away-from-home (AFH) meals and their impact on health. In this study, we evaluated the association between the frequency of AFH meals and the prevalence of allergic rhinitis (AR) and chronic rhinosinusitis (CRS). In this cross-sectional study, the Korean National Health and Nutrition Examination Survey was reviewed from 2010 to 2019. The frequency of AFH meals was assessed as how often the respondents ate AFH meals in an average week. Those who ate AFH meals less than once per week were designated as group 1, one to four times as group 2, and five times or more as group 3. The diagnoses of AR and CRS were evaluated, and symptoms, endoscopic findings, and serum immunoglobulin E (IgE) levels were assessed. Logistic regression analyses were performed. A total of 48,515 participants were eligible. In multivariate logistic regression analysis for AR, when compared to group 1, the odds ratios (ORs) for AR in participants of group 2 (OR = 1.226, 95% confidence interval [CI] = 1.136–1.324) and group 3 (OR = 1.227, 95% CI = 1.126–1.337) were significantly higher (*p* < 0.0001). For CRS, group 2 (OR = 1.139, 95% CI = 1.029–1.260) and group 3 (OR = 1.210, 95% CI = 1.078–1.358) showed a significantly higher risk than group 1 (*p* = 0.0044). Individuals who consume AFH meals frequently might suffer less from AR or CRS if they change their dietary habits and prepare meals more often at home.

## 1. Introduction

According to the World Health Organization, since the coronavirus disease 2019 (COVID-19) pandemic began, approximately 650 million people have been infected, causing 7 million deaths. To control the spread of the virus, a large number of individuals have been subjected to social distancing and lockdowns. When in quarantine, lifestyle changes are inevitable, including dietary habits. It was reported that more people have ordered away-from-home (AFH) meals using food delivery applications on smartphones [1], and that nearly 30% of those delivery application users ordered more frequently during lockdowns [2]. In the US, the expense on food delivery increased when COVID-19 began to spread, even though total household spending decreased in that period [3]. These reports indicate that more people were likely to consume AFH meals rather than to prepare food themselves.

Meanwhile, allergic rhinitis (AR) and chronic rhinosinusitis (CRS) are common upper-airway inflammatory diseases that cause significant impairments in the patients’ quality of life, along with socioeconomic costs for the control of the diseases [4,5,6,7]. The causes of inflammatory sinonasal diseases are multifactorial, and several studies have reported that lifestyle factors, including dietary habits, are associated with the prevalence of these diseases [8,9,10]. Moreover, although comorbid AR or CRS is not speculated to be a risk factor for SARS-CoV-2 infection [11,12], it was reported that the severity of AR symptoms increased during the COVID-19 lockdown period, when individuals used more food delivery services [13]. It could be speculated that lifestyle changes such as AFH meals could have impacted individuals with inflammatory sinonasal diseases.

In this cross-sectional study, we aimed to investigate the association between AFH meals and AR or CRS by analyzing the data of the South Korean population. In addition, disease-specific features, including subjective symptoms of AR or CRS, nasal endoscopic examination findings recorded by otorhinolaryngologists, and serum immunoglobulin E (IgE) levels, were studied in relation to the frequency of AFH meals.

## 2. Materials and Methods

### 2.1. Study Population

This cross-sectional study used data from the Korean National Health and Nutrition Examination Survey (KNHANES), a population-based study organized by the Korean Center for Disease Control and Prevention (K-CDC). This nationwide survey has been conducted annually since 1998 to assess the health and nutritional status of the general Korean population. It is designed to represent the general South Korean population using a multistage, clustered, and stratified random sampling method, and approximately 10,000 individuals from 4000 households are enrolled every year. Trained interviewers and physicians collect data from each participant, including health interviews, physical examinations, and nutritional reports. The K-CDC posts data every year on the KNHANES website, which is used by academic researchers and administrative agencies.

The Korean Society of Otorhinolaryngology-Head and Neck Surgery participated in this survey from 2008 to 2012; 150 residents from the otorhinolaryngology departments of 47 training hospitals conducted thorough medical interviews and physical and endoscopic examinations in a standardized manner. We included data from adult participants (aged ≥18 years) enrolled in the KNHANES from 2010 to 2019. Data from the participants enrolled before 2010 were not used, because the questionnaire asking about the frequency of AFH meals was different before 2010.

In the 10-year enrollment of participants from KNHANES 2010 to 2019, a total of 80,861 participants were enrolled (Figure 1). Participants under 18 years of age and those with missing data for the designated variables were excluded. A total of 61 participants who were found to have nasal cavity tumors on endoscopy were excluded. Consequently, 48,515 participants were eligible for inclusion in this study. Among them, 6642 were diagnosed with AR and 3144 with CRS.

### 2.2. Diagnosis of Inflammatory Sinonasal Diseases and Assessment of Disease-Related Findings

We used specific questionnaires included in the KNHANES regarding inflammatory sinonasal diseases, such as AR and CRS. Each participant was asked by a physician whether they had been diagnosed with AR or CRS. Additionally, nasal symptoms and endoscopic findings related to AR or CRS were recorded, which were composed in accordance with the Allergic Rhinitis and its Impact on Asthma (ARIA) and the European Position Paper on Rhinosinusitis and Nasal Polyps (EPOS) guidelines [14,15]. Specifically, as for ARIA classifications of AR, participants with AR were asked if their rhinitis symptoms were perennial or seasonal, persistent (>4 days a week and >4 consecutive weeks) or intermittent (<4 days a week or <4 consecutive weeks), and mild (not troublesome for sleep, daily activities, school, or work) or moderate/severe (troublesome).

ImmunoCAP 100 (Thermo Fisher Scientific, Uppsala, Sweden) was used to measure serum total immunoglobulin E (IgE) and specific IgE for house dust mites (*D. farinae*), cockroaches, and dogs in approximately 10% of the total participants of 2010. The cutoff values for serum total and specific IgE levels were 100 kU/L and 0.35 kU/L, respectively. For participants with CRS, symptoms such as nasal obstruction, nasal discharge (anterior or posterior), facial pain or pressure, and reduced sense of smell for 12 weeks or more were assessed according to the EPOS guidelines. Furthermore, endoscopic findings such as pale mucosa and watery rhinorrhea for AR and middle meatus discharge and nasal polyps for CRS in each participant were collected from 2010 to 2012.

### 2.3. Measurement of Frequency of Away-From-Home (AFH) Meals

The frequency of AFH meals was measured by asking each participant how many AFH meals they had consumed on average during the past year. They were instructed to include delivery and take-out foods. The choice for answers was given as seven multiple choices: less than once a month, once to thrice a month, once to twice a week, three to four times a week, five to six times a week, once every day, or more than twice a day. For statistical analyses, the answers from the participants were categorized into one of three groups: group 1, less than once a week; group 2, one to four times a week; and group 3, five times a week or more (Figure 2). The prevalence of AR and CRS was analyzed and compared among the three groups.

### 2.4. Assessment of Covariates

The baseline characteristics of each participant were collected, including age, sex, residence, education level, household income, occupation, drinking and smoking status, and obesity. Residence was categorized as either urban or rural, and education levels were divided into four categories based on the academic level of graduation. Household income was divided into four quartiles, and occupations were categorized into seven types of labor. Drinking status was divided into non-drinkers, who had never consumed any alcohol or consumed less than once a month, and drinkers, who consumed once a month or more often. Smoking status was divided into non- or ex-smokers, and current smokers. Obesity was divided based on body mass index (BMI) according to the South Korean obesity classification standards: low weight (<18.5 kg/m^2^), normal (≥18.5 kg/m^2^, <25 kg/m^2^), or obese (≥25 kg/m^2^).

### 2.5. Statistical Analyses

Statistical Analysis System version 9.4 (SAS Institute, Inc., Cary, NC, USA) was used for statistical analyses. The baseline characteristics of the participants were investigated using Student’s *t*-test for continuous variables, such as age, and the chi-squared test for categorical variables. To evaluate the association between the frequency of AFH meals and AR or CRS, univariable and multivariable logistic regression analyses were performed. In each AFH meals group, the odds ratio (OR) and 95% confidence interval (CI) for AR or CRS were measured, and the statistical significance level was set at *p* < 0.05. In the multivariable logistic regression analysis, covariates with significant effects (*p* < 0.25) in the univariate logistic regression analysis were adjusted as confounding factors to avoid omitting their potentially relevant effects. Among the participants with AR, subjective symptoms, serum IgE levels, and endoscopic findings were analyzed according to the frequency of AFH meals. Similarly, for the participants with CRS, subjective symptoms and endoscopic findings were analyzed according to the frequency of AFH meals to find any statistical differences between the AFH meals groups.

### 2.6. Ethical Considerations

Informed consent was obtained from each participant, and the Institutional Review Board of the KCDC approved the investigation of the KNHANES (2010-02CON-21-C, 2011-02CON-06-C, 2012-01EXP-01-2C, 2013-07CON-03-4C, 2013-12EXP-03-5C, 2018-01-03-P-A, and 2018-01-03-C-A).

## 3. Results

### 3.1. Baseline Characteristics

When the participants with AR and the control group were compared, the prevalence of AR was approximately 13.7%, the AR group was younger (mean 43.6 years) than the control group (52.8 years), and the proportion of men (35.09%) was smaller than in the control group (42.18%) (Table 1). Other variables, such as residence, education level, household income, occupation, drinking, smoking, and obesity, also showed statistical differences between the AR and control groups.

In the evaluation of CRS, the prevalence of CRS was approximately 6.5%. The CRS group was younger (mean 48.7 years) than the control group (51.8 years), and 43.03% were men in the CRS group, compared to 41.08% in the control group (Table 2). Educational levels, household income, and occupations were also statistically different between the two groups.

As described in Table 1 and Table 2, we measured the proportion of each AFH meals group according to the AR and control groups or the CRS and control groups (Figure 3). Among the participants with AR, approximately 42.6% were in group 3 (people who consumed AFH meals five times a week or more), while only 19.7% were in group 1 (AFH meals less than once a week). In the control group, the distribution among groups 1, 2, and 3 was even (approximately 33%). Similarly, in the CRS group, the proportion of group 3 was 39.4%, and that of group 1 was 25.9%, while in the control participants the distribution was almost even (31.8 to 34.4%). The distribution patterns of the AFH meal frequency groups between the AR and control groups, and between the CRS and control groups, were both different and statistically significant (*p* < 0.0001) (Table 1 and Table 2).

### 3.2. Association Between Inflammatory Sinonasal Diseases and Frequency of AFH Meals

When the 48,515 participants were evaluated for their risk of AR, the frequency of AFH meals showed a significant association with the diagnosis of AR (Table 3). Univariable logistic regression analysis showed that when group 1 was set as the reference, the crude ORs for the diagnosis of AR were 1.915 (95% CI = 1.784–2.055, *p* < 0.0001) in group 2 and 2.141 (95% CI = 1.997–2.295, *p* < 0.0001) in group 3 (*p* < 0.0001). In the multivariable logistic regression analysis, when adjusted for the other variables with significant effects, the adjusted ORs for the diagnosis of AR were 1.226 (95% CI = 1.136–1.324, *p* < 0.0001) in group 2 and 1.227 (95% CI = 1.126–1.337, *p* < 0.0001) in group 3 (*p* < 0.0001).

In the evaluation of CRS among the total participants, the frequency of AFH meals again showed a statistically significant association with the diagnosis of CRS (Table 4). Univariable logistic regression analysis for the diagnosis of CRS showed that when group 1 was set as the reference, the crude ORs for CRS were 1.258 (95% CI = 1.146–1.381, *p* < 0.0001) in group 2 and 1.399 (95% CI = 1.277–1.532, *p* < 0.0001) in group 3. Multivariate regression analysis showed that the adjusted ORs for CRS were 1.139 (95% CI = 1.029–1.260, *p* = 0.0119) in group 2 and 1.210 (95% CI = 1.078–1.358, *p* = 0.0013) in group 3 (*p* = 0.0044).

### 3.3. Serum IgE Levels and Endoscopic Findings in Participants with AR

Among participants with AR, we analyzed the association between the frequency of AFH meals and each clinical finding of AR. First, for the 1442 participants with AR enrolled from 2010 to 2012, we investigated whether the ARIA classifications of AR were associated with the frequency of AFH meals. In the multivariable logistic regression analysis, however, whether the rhinitis symptoms were perennial or seasonal, persistent or intermittent, and mild or moderate/severe, none showed any significant association with the frequency of AFH meals.

Next, among the 699 participants with AR in 2010 who were tested with ImmunoCAP 100, the positive serum-specific IgE result for house dust mites was significantly associated with the frequency of AFH meals (Table 5). When compared to group 1, the adjusted OR for the positive serum-specific IgE result for house dust mites was 1.324 (95% CI = 0.760–2.307, *p* = 0.3215) in group 2 and 2.007 (95% CI = 1.198–3.363, *p* = 0.0082) in group 3 (*p* = 0.0173). The serum total IgE and specific IgE levels for cockroaches and dogs did not show statistically significant differences in the AFH meal frequency groups.

Finally, among the 1970 participants with AR enrolled from 2010 to 2012 who had undergone nasal endoscopy examinations by otorhinolaryngology residents, the endoscopic finding of watery rhinorrhea was significantly associated with the frequency of AFH meals (Table 5). Compared to group 1, the adjusted OR for having pale mucosa on endoscopy was 1.121 (95% CI = 0.871–1.442, *p* = 0.3743) in group 2 and 1.192 (95% CI = 0.931–1.526, *p* = 0.1648) in group 3 (*p* = 0.3809). However, the adjusted OR for the positive finding of watery rhinorrhea on nasal endoscopy compared to group 1 was 1.033 (95% CI = 0.796–1.341, *p* = 0.8044) in group 2 and 1.319 (95% CI = 1.025–1.697, *p* = 0.0311) in group 3 (*p* = 0.0359).

### 3.4. Symptoms and Endoscopic Findings in Participants with CRS

The 836 participants with CRS enrolled from 2010 to 2012 who had data for the disease-related findings of CRS, such as subjective symptoms and nasal endoscopic findings, in accordance with the EPOS 2020 guidelines, were assessed. In the multivariable logistic regression analysis, among the four symptom criteria for CRS, the association between reduced sense of smell and frequency of AFH meals was statistically significant, with an adjusted OR of 1.717 (95% CI = 1.169–2.522, *p* = 0.0058) for group 2 and 2.164 (95% CI = 1.456–3.217, *p* = 0.0001) for group 3 compared with group 1 (*p* = 0.0004) (Table 6). The frequency of AFH meals did not show statistically significant effects on the other three symptom criteria of CRS: nasal obstruction, nasal discharge, and facial pain. Similarly, among the nasal endoscopic findings, neither middle meatus discharge nor the presence of nasal polyps on nasal endoscopy showed a statistically significant association with the frequency of AFH meals.

## 4. Discussion

To the best of our knowledge, this is the first study to investigate the association between the frequency of AFH meals and inflammatory sinonasal diseases, such as AR and CRS. We analyzed the relationship by obtaining population-based real-world data representing the general population of South Koreans enrolled from 2010 to 2019 and creating logistic regression models [16]. The data not only contain general health characteristics collected with questionnaires from the participants but also feature the clinical examination findings of nasal endoscopies. The otorhinolaryngology residents performed interviews and endoscopic examinations of each participant individually in a validated manner, according to clinical guidelines [15,17,18].

Our finding was that individuals who ate AFH meals more frequently had a higher risk for the diagnosis of both AR and CRS than those who ate AFH meals less frequently. Additionally, among the participants with AR, the endoscopic finding of watery rhinorrhea and the serum-specific IgE level for house dust mites were associated with the frequency of AFH meals. Participants with CRS showed that, among the CRS symptoms, reduced sense of smell was associated with the frequency of AFH meals.

Several studies have investigated the potential clinical relevance of AFH meals in certain diseases. In a recent population-based study using the data of 35,084 participants from the National Health and Nutrition Examination Survey (NHANES), the hazard ratio of all-cause mortality was 1.49 (95% CI 1.05–2.13) for the participants who had AFH meals frequently compared to those who seldom ate AFH meals [19]. Another cross-sectional study that also used data from the NHANES analyzed the association between the frequency of eating AFH meals and biomarkers of chronic diseases [20]. The results showed that the mean BMI was higher and the serum concentrations of micronutrients such as vitamins B, C, D, and E, as well as α-carotene, were decreased among the participants who ate AFH meals more frequently. Moreover, a study that investigated 723 subjects from Minnesota found that AFH meals were associated with being overweight [21].

Although no previous study has analyzed the association between AFH meals and AR or CRS, several factors can be considered to understand the positive relationship found in our study. Individuals who frequently consume AFH meals are deficient in several nutrients, including vitamin D, vitamin E, and folic acid [20]. Studies have reported a relationship between micronutrients and AR, and a meta-analysis concluded that high vitamin D concentration was associated with a lower prevalence of AR in adults, especially in men, and lower aeroallergen sensitization in children [22]. Another meta-analysis concluded that lower vitamin D levels were related to a higher prevalence of AR in children [23]. Regarding vitamin E, a study reported that treatment with vitamin E in an AR mouse model resulted in reduced AR symptoms and T-helper 2 cytokines [24]. Folic acid was also associated with the risk of AR; a shortage of maternal folic acid intake led to an increased risk of respiratory allergy in infants and children [25]. The aforementioned micronutrients are rich in vegetables, fruits, fish, nuts, and vegetable oil, which may not be abundant in AFH meals. Individuals who frequently consume AFH meals might be short of micronutrients, which have immunological regulatory functions, such as Th2 response inhibition [26].

Similar results have been reported in the literature for CRS. In a recent study, primary human nasal epithelial cells from patients with CRS with nasal polyps were collected and treated with lipopolysaccharide, followed by calcitriol, the active form of vitamin D [27]. The results showed that the production of interleukin (IL)-6 was alleviated and the phosphorylation of p65, which represents the nuclear factor kappa-light-chain-enhancer of activated B cells (NF-κB) pathway, was significantly decreased.

Regarding the possible theory of the association between AFH meals and watery rhinorrhea symptoms in patients with AR, an animal study applied topical retinoic acid (vitamin A) to the maxillary sinus mucosa of rabbits [28]. The authors found that the topical vitamin A group showed enhanced regeneration of damaged paranasal mucosa, including submucosal gland loss, compared to the control group without vitamin A application. Sensitization to house dust mites might be related to the frequency of AFH meals owing to the lack of vitamin D. A recent study evaluated the vitamin D status of patients with atopic dermatitis sensitized to house dust mites and found that there was a negative correlation between specific IgE for house dust mites and serum vitamin D levels [29].

This study has some limitations. Due to its cross-sectional design, it is unable to clearly define the causal relationship between AFH meals and AR or CRS. In addition, the types of food or nutrients that the study participants had consumed were not included in the analysis. KNHANES data included the usual types of food that the participants took daily. However, the questionnaire for food type has not been consistently organized during the 10-year survey duration, making it difficult to quantify which foods or nutrients should be taken into account. Thus, we used the frequency of AFH meals as an independent variable. Surveying not only the frequency of AFH meals but the exact categories of foods could further enhance measurement of the individual effects of each micronutrient in AFH meals on AR or CRS.

The results of this study emphasize the possible importance of dietary habits such as AFH meals in patients with AR or CRS. Both inflammatory diseases of the nasal and paranasal sinus mucosa could worsen by consuming AFH meals, and this study might suggest the additional lifestyle modification treatment of eating healthily at home. As AFH meals grow more popular, further research, such as a prospective study regarding what types of food or nutrients affect AR or CRS, may help us to understand the association between AFH meals and AR or CRS.

## 5. Conclusions

The prevalence of AR and CRS was significantly associated with the frequency of AFH meals among the South Korean population. Among the participants with AR, endoscopic findings of watery rhinorrhea and serum IgE for house dust mites were related to the frequency of AFH meals. In the participants with CRS, olfactory dysfunction was associated with AFH meals. The results suggest that the nutritional imbalance associated with frequent AFH meals could affect the condition of individuals with AR or CRS. Therefore, it is feasible that individuals who frequently consume AFH meals might lower the clinical severity of their AR and CRS if they cook and eat at home more often.

## Figures and Tables

**Figure 1 diseases-12-00286-f001:**
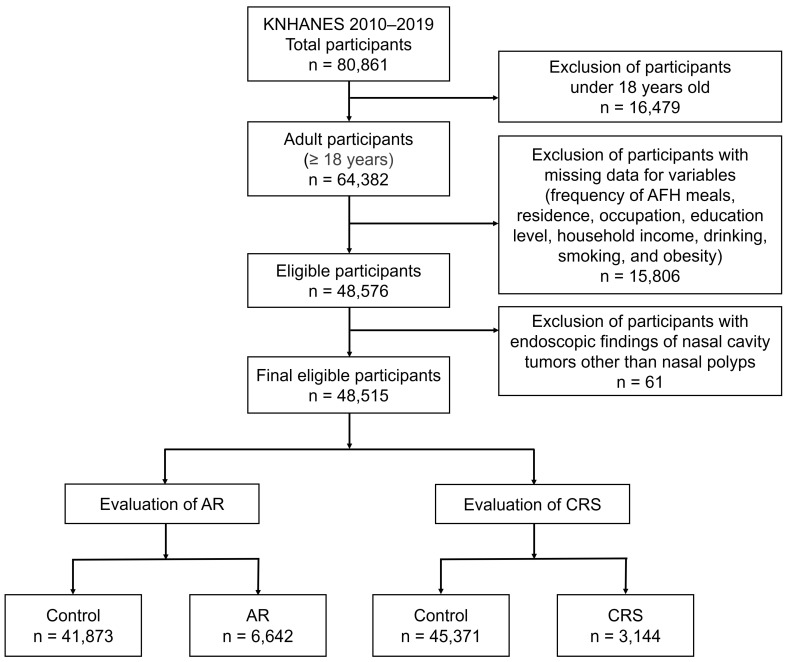
Study participants eligible for evaluation of AR and CRS. KNHANES, Korean National Health and Nutrition Examination Survey; AR, allergic rhinitis; CRS, chronic rhinosinusitis.

**Figure 2 diseases-12-00286-f002:**
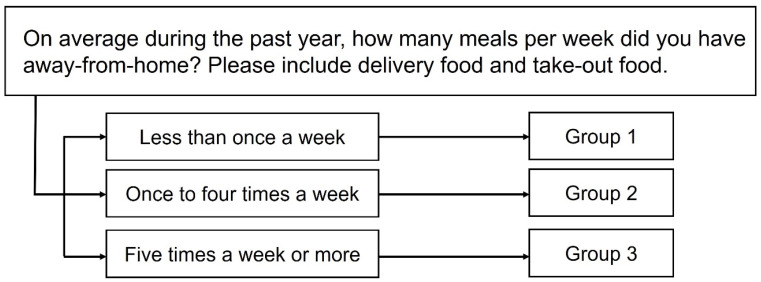
Frequency of away-from-home meals was asked with a questionnaire item. Each participant was categorized into one of three groups according to the frequency of away-from-home meals.

**Figure 3 diseases-12-00286-f003:**
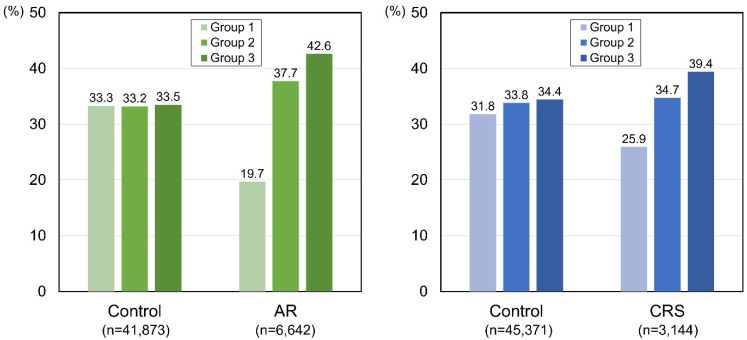
Percentages of groups 1, 2, and 3 among the controls and the participants with AR, and among the controls and the participants with CRS. Group 1, the participants who had AFH meals less than once a week; group 2, one to four times a week; group 3, five times a week or more. AR, allergic rhinitis; CRS, chronic rhinosinusitis.

**Table 1 diseases-12-00286-t001:** Baseline characteristics according to diagnosis of AR.

	Control (n = 41,873, 86.3%)	AR (n = 6642, 13.7%)	*p*-Value
Age, years (mean ± SD)	52.8 ± 16.5	43.6 ± 15.21	**<0.0001**
Sex, n (%)			**<0.0001**
Male	17,660 (42.18)	2331 (35.09)	
Female	24,213 (57.82)	4311 (64.91)	
Residence, n (%)			**<0.0001**
Urban	33,118 (79.09)	5695 (85.74)	
Rural	8755 (20.91)	947 (14.26)	
Education level, n (%)			**<0.0001**
≤Elementary school	10,882 (25.99)	663 (9.98)	
Middle school	4642 (11.09)	497 (7.48)	
High school	13,398 (32)	2405 (36.21)	
≥College or graduate school	12,951 (30.93)	3077 (46.33)	
Household income, n (%)			**<0.0001**
Lower	8895 (21.24)	775 (11.67)	
Lower middle	10,566 (25.23)	1675 (25.22)	
Upper middle	11,019 (26.32)	1998 (30.08)	
Upper	11,393 (27.21)	2194 (33.03)	
Occupation, n (%)			**<0.0001**
Managers, experts, and related workers	4991 (11.92)	1223 (18.41)	
Office workers	3654 (8.73)	821 (12.36)	
Service and sales personnel	4981 (11.9)	930 (14)	
Agricultural, forestry, or fishery skilled workers	2741 (6.55)	141 (2.12)	
Technicians, equipment, and machine operation and assembly workers	3961 (9.46)	468 (7.05)	
Simple labor workers	3941 (9.41)	446 (6.71)	
Unemployed (housewife, student, etc.)	17,604 (42.04)	2613 (39.34)	
Drinking, n (%)			**<0.0001**
None (non-drinker or <1/month)	20,407 (48.74)	2947 (44.37)	
Yes (1/month or more often)	21,466 (51.26)	3695 (55.63)	
Smoking, n (%)			**<0.0001**
None (non-smoker or ex-smoker)	34,495 (82.38)	5703 (85.86)	
Yes (current smoker)	7378 (17.62)	939 (14.14)	
Obesity, n (%)			**<0.0001**
Low weight (BMI < 18.5 kg/m^2^)	1669 (3.99)	352 (5.3)	
Normal (18.5 kg/m^2^ ≤ BMI < 25 kg/m^2^)	26,059 (62.23)	4411 (66.41)	
Obese (25 kg/m^2^ ≤ BMI)	14,145 (33.78)	1879 (28.29)	
Frequency of AFH meals, n (%)			**<0.0001**
Group 1 (<1/week)	13,926 (33.26)	1310 (19.72)	
Group 2 (1–4/week)	13,908 (33.21)	2505 (37.71)	
Group 3 (≥5/week)	14,039 (33.53)	2827 (42.56)	

*p*-Values with statistical significance are in bold. BMI values were divided according to South Korean obesity classification values. AR, allergic rhinitis; SD, standard deviation; BMI, body mass index; AFH, away-from-home.

**Table 2 diseases-12-00286-t002:** Baseline characteristics according to diagnosis of CRS.

	Control (n = 45,371, 93.5%)	CRS (n = 3144, 6.5%)	*p*-Value
Age, years (mean ± SD)	51.8 ± 16.6	48.7 ± 16.9	**<0.0001**
Sex, n (%)			**0.0312**
Male	18,638 (41.08)	1353 (43.03)	
Female	26,733 (58.92)	1791 (56.97)	
Residence, n (%)			0.2738
Urban	36,274 (79.95)	2539 (80.76)	
Rural	9097 (20.05)	605 (19.24)	
Education level, n (%)			**<0.0001**
≤Elementary school	10,929 (24.09)	616 (19.59)	
Middle school	4853 (10.7)	286 (9.1)	
High school	14,841 (32.71)	962 (30.6)	
≥College or graduate school	14,748 (32.51)	1280 (40.71)	
Household income, n (%)			**0.0282**
Lower	9088 (20.03)	582 (18.51)	
Lower middle	11,459 (25.26)	782 (24.87)	
Upper middle	12,184 (26.85)	833 (26.49)	
Upper	12,640 (27.86)	947 (30.12)	
Occupation, n (%)			**<0.0001**
Managers, experts, and related workers	5696 (12.55)	518 (16.48)	
Office workers	4163 (9.18)	312 (9.92)	
Service and sales personnel	5542 (12.21)	369 (11.74)	
Agricultural, forestry, or fishery skilled workers	2733 (6.02)	149 (4.74)	
Technicians, equipment, and machine operation and assembly workers	4173 (9.2)	256 (8.14)	
Simple labor workers	4157 (9.16)	230 (7.32)	
Unemployed (housewife, student, etc.)	18,907 (41.67)	1310 (41.67)	
Drinking, n (%)			0.5685
None (non-drinker or <1/month)	21,856 (48.17)	1498 (47.65)	
Yes (1/month or more often)	23,515 (51.83)	1646 (52.35)	
Smoking, n (%)			0.3789
None (non-smoker or ex-smoker)	37,575 (82.82)	2623 (83.43)	
Yes (current smoker)	7796 (17.18)	521 (16.57)	
Obesity, n (%)			0.8460
Low weight (BMI < 18.5 kg/m^2^)	1885 (4.15)	136 (4.33)	
Normal (18.5 kg/m^2^ ≤ BMI < 25 kg/m^2^)	28,490 (62.79)	1980 (62.98)	
Obese (25 kg/m^2^ ≤ BMI)	14,996 (33.05)	1028 (32.7)	
Frequency of AFH meals, n (%)			**<0.0001**
Group 1 (<1/week)	14,420 (31.78)	816 (25.95)	
Group 2 (1–4/week)	15,322 (33.77)	1091 (34.7)	
Group 3 (≥5/week)	15,629 (34.45)	1237 (39.34)	

*p*-Values with statistical significance are in bold. CRS, chronic rhinosinusitis; SD, standard deviation; BMI, body mass index; AFH, away-from-home.

**Table 3 diseases-12-00286-t003:** Logistic regression analysis of the association between frequency of AFH meals and diagnosis of AR.

	Crude OR (95% CI)	*p*-Value	Adjusted OR * (95% CI)	*p*-Value
Frequency of AFH meals		**<0.0001**		**<0.0001**
Group 1 (<1/week)	Ref		Ref	
Group 2 (1–4/week)	1.915 (1.784–2.055)	**<0.0001**	1.226 (1.136–1.324)	**<0.0001**
Group 3 (≥5/week)	2.141 (1.997–2.295)	**<0.0001**	1.227 (1.126–1.337)	**<0.0001**

* Adjusted for age, sex, residency, education level, household income, occupation, drinking, smoking, obesity, and frequency of AFH meals; *p*-values with statistical significance are in bold. AFH, away-from-home; AR, allergic rhinitis; OR, odds ratio; CI, confidence interval; Ref, reference.

**Table 4 diseases-12-00286-t004:** Logistic regression analysis of the association between frequency of AFH meals and diagnosis of CRS.

	Crude OR (95% CI)	*p*-Value	Adjusted OR * (95% CI)	*p*-Value
Frequency of AFH meals		<0.0001		**0.0044**
Group 1 (<1/week)	Ref		Ref	
Group 2 (1–4/week)	1.258 (1.146–1.381)	<0.0001	1.139 (1.029–1.260)	**0.0119**
Group 3 (≥5/week)	1.399 (1.277–1.532)	<0.0001	1.210 (1.078–1.358)	**0.0013**

* Adjusted for age, sex, residency, education level, household income, occupation, and frequency of AFH meals; *p*-values with statistical significance are in bold. AFH, away-from-home; CRS, chronic rhinosinusitis; OR, odds ratio; CI, confidence interval; Ref, reference.

**Table 5 diseases-12-00286-t005:** Logistic regression analysis of the association between frequency of AFH meals and serum IgE levels and nasal endoscopic findings among participants with AR.

	Adjusted OR (95% CI)	*p*-Value
Total IgE positive		0.1672
Group 1 (<1/week)	1 (reference)	
Group 2 (1–4/week)	0.900 (0.506–1.602)	0.7198
Group 3 (≥5/week)	1.394 (0.826–2.355)	0.2137
Specific IgE positive for house dust mites		**0.0173**
Group 1 (<1/week)	1 (reference)	
Group 2 (1–4/week)	1.324 (0.760–2.307)	0.3215
Group 3 (≥5/week)	2.007 (1.198–3.363)	**0.0082**
Specific IgE positive for cockroaches		0.0661
Group 1 (<1/week)	1 (reference)	
Group 2 (1–4/week)	0.719 (0.279–1.852)	0.4946
Group 3 (≥5/week)	1.726 (0.786–3.789)	0.1739
Specific IgE positive for dogs		0.1085
Group 1 (<1/week)	1 (reference)	
Group 2 (1–4/week)	0.725 (0.179–2.941)	0.6531
Group 3 (≥5/week)	2.151 (0.697–6.642)	0.1831
Pale mucosa in nasal endoscopy		0.3809
Group 1 (<1/week)	1 (reference)	
Group 2 (1–4/week)	1.121 (0.871–1.442)	0.3743
Group 3 (≥5/week)	1.192 (0.931–1.526)	0.1648
Watery rhinorrhea in nasal endoscopy		**0.0359**
Group 1 (<1/week)	1 (reference)	
Group 2 (1–4/week)	1.033 (0.796–1.341)	0.8044
Group 3 (≥5/week)	1.319 (1.025–1.697)	**0.0311**

*p*-Values with statistical significance are in bold. AFH, away-from-home; IgE, immunoglobulin E; AR, allergic rhinitis; OR, odds ratio; CI, confidence interval.

**Table 6 diseases-12-00286-t006:** Logistic regression analysis of the association between frequency of AFH meals and symptoms of CRS and nasal endoscopic findings according to the EPOS 2020 guidelines among participants with CRS.

	Adjusted OR (95% CI)	*p*-Value
Nasal obstruction for 12 weeks or more		0.7254
Group 1 (<1/week)	1 (reference)	
Group 2 (1–4/week)	1.004 (0.734–1.373)	0.9797
Group 3 (≥5/week)	0.909 (0.671–1.233)	0.5399
Nasal discharge (anterior or posterior) for 12 weeks or more		0.3851
Group 1 (<1/week)	1 (reference)	
Group 2 (1–4/week)	1.181 (0.865–1.611)	0.2943
Group 3 (≥5/week)	1.234 (0.908–1.678)	0.1797
Facial pain or pressure for 12 weeks or more		0.6560
Group 1 (<1/week)	1 (reference)	
Group 2 (1–4/week)	1.421 (0.662–3.051)	0.3669
Group 3 (≥5/week)	1.267 (0.610–2.633)	0.5258
Reduced sense of smell for 12 weeks or more		**0.0004**
Group 1 (<1/week)	1 (reference)	
Group 2 (1–4/week)	1.717 (1.169–2.522)	**0.0058**
Group 3 (≥5/week)	2.164 (1.456–3.217)	**0.0001**
Middle meatus discharge on nasal endoscopy		0.6982
Group 1 (<1/week)	1 (reference)	
Group 2 (1–4/week)	0.832 (0.470–1.472)	0.5275
Group 3 (≥5/week)	1.022 (0.595–1.755	0.9364
Nasal polyp on nasal endoscopy		0.6738
Group 1 (<1/week)	1 (reference)	
Group 2 (1–4/week)	0.734 (0.371–1.455)	0.3761
Group 3 (≥5/week)	0.830 (0.431–1.597)	0.5767

*p*-Values with statistical significance are in bold. AFH, away-from-home; CRS, chronic rhinosinusitis; EPOS, European Position Paper on Rhinosinusitis and Nasal Polyps; OR odds ratio; CI, confidence interval.

## Data Availability

The KNHANES data are open to the public; therefore, any researcher can obtain the raw data by requesting them on the website (https://knhanes.kdca.go.kr/knhanes/eng/index.do, accessed on 1 August 2021).

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
