# Peer review of "Association Between Frequency of Away-From-Home Meals and Prevalence of Inflammatory Sinonasal Diseases: A Population-Based Cross-Sectional Study"

_diseases, 2024, doi:10.3390/diseases12110286_

Round 1

Reviewer 1 Report

Comments and Suggestions for Authors

Hi dear Editorial board and the respected authors

This article "Frequency of Away-from-Home Meals Is Associated with Prevalence of Inflammatory Sinonasal Diseases: A Population-Based Cross-Sectional Study” was revised and has a novelty and I recommend it for publication after consideration of the following comments.

Title: If you can rewrite and make it more interesting for readers. I propose: “Association Between Away-from-Home Meal Frequency and Inflammatory Sinonasal Diseases: A Cross-Sectional Study”.

Abstract:

·         How do you account for any confounding factors that might affect both the frequency of out-of-home meals and the incidence of inflammatory sinonasal diseases? Could other dietary or lifestyle variables influence the observed associations?

·         Given that the study is focused on South Korean individuals, how do you ensure that the findings apply to other populations or cultural contexts with varying eating patterns and health profiles?

·         What particular criteria were utilized to identify meals as "away-from-home"? How did you account for changes in the nutritional content of these meals, and how would this variability affect the findings on AR and CRS prevalence?

Keywords: Please have the minimum 5 keywords.

Introduction: Please extend the text of Introduction.

You can refer to the following articles:

·         Journal of Food Safety 40 (6), e12853

·         Food science & nutrition 8 (10), 5228-5237

Results:

·         Figure 3: The alphabetical statistical letters for the means should all be modified such that the greatest number has the letter a and as the numbers go lower, letters b, c etc.

Discussion:

Discussion text must grammar improve and in some cases it is very weak and maybe there is no discussion at all.

·         Given your study's cross-sectional nature, how do you plan to demonstrate a causal association between the frequency of away-from-home (AFH) meals and inflammatory sinonasal disorders (AR and CRS) in future research? What longitudinal techniques may be used to solve this limitation?

·         You state that some micronutrient deficits (such as vitamin D, E, and folic acid) are linked to AFH meals. How do you intend to measure or evaluate the precise nutritional composition of AFH meals in future research to better understand their effect on AR and CRS?

·         How did you adjust for any potential confounding factors that may affect the prevalence of AR and CRS, such as environmental exposures, socioeconomic level, or pre-existing health?

Conclusions:

Conclusion is not available. Please include the best comprehensive conclusion.

References: It is OK.

Reviewer 2 Report

Comments and Suggestions for Authors

Thank you for the opportunity to review the paper titled ‘Frequency of Away-from-Home Meals Is Associated with Prevalence of Inflammatory Sinonasal Diseases: A Population-Based Cross-Sectional Study’. The study that elucidates the association of AFH with inflammatory sinonasal diseases is interesting and original. However, the following revisions are needed:  

1. The abstract is too long and needs to be concisely summarized.

 2. In the Methods section, it seems that studies utilizing big data require exemption from institutional IRB review. Currently, only K-CDC has IRB approval.  

3. It would be more systematic to move Figure 2 in the Results section to the Methods section Participants section.

 4. The Discussion section lacks explanation of the clinical significance of this study and requires supplementation. In addition, the final conclusion of this study should be added.

Reviewer 3 Report

Comments and Suggestions for Authors

Comments are attached.

Reviewer 4 Report

Comments and Suggestions for Authors

Please ensure the format of MDPI for the abstract.

Authors need to present the research question by thorough literature review. The background of this work is very weak. Authors need to present the research gap more logically.

Authors also need to present the explcation of data collection procedure.

Authors need to present the statistical tools for the data analysis presenting the threshold using reference.

Authors additionally need to improve the implication part of this work more. What could become theoretical and practical  value of this work?

Round 2

Reviewer 3 Report

Comments and Suggestions for Authors

no more comments